


# Improved BEC SMOS Arctic Sea Surface Salinity product v3.1

Justino Martínez[1], Carolina Gabarró[1], Antonio Turiel[1], Verónica González-Gambau[1], Marta Umbert[1], Nina Hoareau[1], Cristina González-Haro[1], Estrella Olmedo[1], Manuel Arias[2], Rafael Catany[2], Laurent Bertino[3], Roshin Raj[3], Jiping Xie[3], Roberto Sabia[4], and Diego Fernandez[5]

[1] Barcelona Expert Center (BEC) and Institute of Marine Sciences (ICM), CSIC, P. Marítim de la Barceloneta, 37-49, 08003 Barcelona, Spain
[2]ARGANS, Derriford, PL6 8BX Plymouth, U.K.
[3]Nansen Environmental and Remote Sensing Center- NERSC, Jahnebakken 3, N-5007 Bergen, Norway
[4]Telespazio-Vega UK Ltd., for ESA-ESRIN, Largo Galileo Galilei 1, 00044 Frascati, Italy
[5]ESA-ESRIN, Largo Galileo Galilei 1, 00044 Frascati, Italy

**Correspondence:** Carolina Gabarro (cgabarro@icm.csic.es)

**Abstract.** Measuring salinity from space is challenging since the sensitivity of the brightness temperature ($TB$) to sea surface salinity (SSS) is low (about $0.5K/psu$), while the SSS range in the open ocean is narrow (about 5 psu, if river discharge areas are not considered). This translates into a high accuracy requirement of the radiometer (about 2-3 K). Moreover, the sensitivity of the $TB$ to SSS at cold waters is even lower ($0.3K/psu$), making the retrieval of the SSS in the cold waters even more challenging. Due to this limitation, ESA launched a specific initiative in 2019, the Arctic+Salinity project (AO/1-9158/18/I-BG), to produce an enhanced Arctic SSS product with better quality and resolution than the available products. This paper presents the methodologies used to produce the new enhanced Arctic SMOS SSS product (Martínez *et al.*, 2020) . The product consists of 9-day averaged maps in an EASE 2.0 grid of 25 km. The product is freely distributed from the Barcelona Expert Center (BEC, http://bec.icm.csic.es/) with the DOI number: 10.20350/digitalCSIC/12620. The major change in this new product is its improvement of the effective spatial resolution that permits better monitoring of the mesoscale structures (larger than 50 Km), which benefits the river discharges monitoring.

## 1 Introduction

Changes in the Arctic Ocean freshwater distribution may be linked to changes in the thermohaline circulation, which in turn may have implications for the global climate (Manabe, 1995). Thus, it is critical to understand the mechanisms of freshwater exchanges between the Arctic and the global ocean. Thus, it is critical to understand the mechanisms of freshwater exchanges between the Arctic and the global ocean. However, the acquisition of continuous series of salinity measurements at high latitudes is a difficult task. The Arctic is a region with extreme weather conditions and sea ice forces are strong enough to destroy the in situ measurements infrastructures (like Argo floats, moorings, or gliders). The number of in situ surface salinity measurements is, therefore, very scarce, and especially in the central Arctic Ocean.

The use of L-band radiometry to fill the observational salinity gaps at high latitudes plays a key role in better monitoring the observed changes in the freshwater fluxes. In 2009, the ESA SMOS (Soil Moisture Ocean Salinity) satellite mission was





launched (Kerr *et al.*, 2010). It was the first satellite carrying an L-band radiometer enabling the measurement of the ocean sea surface salinity.

The SMOS standard SSS retrieval algorithm (Font *et al.*, 2010; Mecklenburg *et al.*, 2009; Olmedo *et al.*, 2021), as well as the algorithms used for SSS retrieval from Aquarius and SMAP data (Tang *et al.*, 2017, 2020), provide in general good estimates of SSS in the open ocean and within the tropical and mid-latitudes (Reul *et al.*, 2020).

However, SSS retrievals from the current operating L-band radiometer satellites present serious problems at high latitudes:

– Low sensitivity of Brightness Temperatures (TB) to salinity in cold waters: Whilst L-band frequency is the region of the electromagnetic spectrum offering the most sensitivity to salinity variations, it decreases rapidly in cold waters. As shown in Yueh *et al.* (2001), such sensitivity drops from 0.5 to 0.3 K/psu, when Sea Surface Temperature (SST) decreases from 15 to 5°C. Therefore, the errors of the SSS in cold waters are larger than in temperate seas.

– Land-sea contamination (LSC) and ice–sea contamination (ISC): Sharp transititons of $TB$ values between land and sea, or ice and sea, induce contamination of the signal which is especially important (both in amplitude and spatial range) in the case of SMOS, due to its interferometric nature. Despite this instrumental characteristic, it is also present in SMAP and its predecessor, Aquarius. In the case of SMOS, this type of contamination has an impact on ocean observations very far from the coast and the ice.

– Lack of in situ measurements: The limited number of in situ measurements of SSS in the Arctic is a major imitations, for the validation, since measurements are not evenly distributed, so that some regions have a clear lack of them.

In the framework of the ESA project Arctic+Salinity (AO/1-9158/18/I-BG), 9 years (2011-2019) of an enhanced SMOS SSS product have been produced. BEC distributes Level 2 maps and Level 3 maps of 3, 9, and 18 days in an EASE 2.0 grid of 25 km. The major changes in the algorithms have been focused on improving the effective spatial and temporal resolution of the product, allowing better monitoring of the mesoscale structures and the river discharges. The algorithms used for the generation of this new product are detailed in the Algorithm Theoretical Baseline Document (ATBD) of the Arctic+Salinity product (Martínez *et al.*, 2020).

This paper describes the data sets (section 1.1) used for the generation of the product and its validation, the algorithms developed to generate the new product (section 2) and the validation performed to assess the quality of the product, namely: i) comparison with in situ measurements, ii) spectral analysis, and iii) error characterization by using triple collocation with SMAP data (section 3).

## 1.1 Data sets

### 1.1.1 SMOS Brightness Temperatures

The computation of $TB$ starts from the ESA Level 1B v620 product. This data set is is freely available prior registration at: https://earth.esa.int/eogateway/missions/smos/data.

L1B product contains $TB$ Fourier components arranged in a time-ordered way according to the integration time.



### 1.1.2 Auxiliary data used in the salinity retrieval

**Geophysical variables from the European Centre for Medium-Range Weather Forecasts (ECMWF)**

The auxiliary information is provided by ECMWF (Sabater & De Rosnay, 2010) collocated in time and space with each SMOS orbit. The data is provided in the Icosahedral Snyder Equal Area 4H9 (ISEA 4H9) grid (Matos *et al.*, 2004). We use a nearest-neighbor interpolation to get the values in the custom $TB$ grid.

**Sea ice concentration from Ocean and Sea Ice Satellite Application Facility (OSISAF)**

Sea ice concentration (SIC) product from OSISAF is used to discard the grid points contaminated by sea ice. We use SIC product version OSI-450 and OSI-430-b (Lavergne *et al.*, 2019) developed and processed in the context of the OSISAF (http://www.osi-saf.org/)) of the European Organisation for the Exploitation of Meteorological Satellites (EUMETSAT) and the Climate Change Initiative (CCI) Programme of the European Space Agency (ESA). The grid of these products is the EASE-Grid 2.0 with a spatial resolution of 25 km.

### 1.1.3 Annual SSS and SST climatology

The World Ocean Atlas 2018 (WOA 2018 - A5B7) annual SSS and SST climatologies at $0.25° \times 0.25°$ spatial resolution (Zweng *et al.*, 2018) are used in the inversion algorithm. The SSS and SST are converted to $TB$ using the Meissner and Wentz dielectric constant model (Meissner & Wentz, 2004; Meissner *et al.*, 2018) and considered as the reference value to perform the spatial bias correction of the measured $TB$. WOA 2018 - A5B7 (generated from measurements of the 2005-2017 period) has been used to be consistent as much as possible with the SMOS life period.

### 1.1.4 Global Ocean Forecasting System

The Global Ocean Forecasting System (GOFS) 3.1 (HYbrid Coordinate Ocean Model (HYCOM) + Navy Coupled Ocean Data Assimilation (NCODA)) sea surface salinity product is used as the reference to perform the temporal correction. The spatial resolution of this product is $0.08°$ in longitude and $0.04°$ in latitude for polar regions and corresponds to the GLBv0.08 grid (Cummings, 2005; Cummings & Smedstad, 2013). The data can be downloaded from https://www.hycom.org/data/glbv0pt08.

### 1.1.5 ARGO dataset

ARGO floats data (Argo (2018)) is commonly used for the validation of SSS from SMOS and SMAP data (Tang *et al.*, 2017; Olmedo *et al.*, 2021). However, this is complicated in the Arctic region. ARGO data is very scarce and Argo profilers are concentrated in the Atlantic region (due to bathymetry and geographical restriction of ocean circulation), providing a biased sample of the mean SSS in the Arctic. There is a lack of Argo profilers in the Bering, Beaufort, East Siberian, Laptev, Kara, Barents, and North seas and also in the Hudson and Baffin bays (see figure 7).

The following quality control over the values of Argo SSS is used: The cut-off depth for Argo profiles is taken between 5 and 10 m. Profiles included in the grey list (i.e., floats which may have problems with one or more sensors) are discarded. Argo float profiles with anomalies larger than $10°C$ in temperature or 5 psu in salinity when compared to WOA 2013 are discarded.





Finally, only profiles having a temperature between -2.5 and 40°C and salinity between 2 and 41 psu close to the surface are used.

### 1.1.6 TARA OCEANS (2009-2013) Expedition dataset

The TARA OCEANS (2009-2013) Expedition collected SST and SSS at 3 m depth during the whole cruise between 2009 and 2013 all around the world thanks to the thermosalinograph (TSG) Seabird SB45 and a temperature sensor (SBE38) that was

onboard its research vessel, TARA. The Arctic Ocean data corresponds to the period of June to October 2013. Hereafter, this dataset is referred to as TARA SSS (Reverdin *et al.*, 2014).

### 1.1.7 JPL SMAP data set

Level 3 SSS maps, version 4.2 provided by Jet Propulsion Laboratory (JPL) (from https://podaac-opendap.jpl.nasa.gov/opendap/allData/smap/L3/JPL/V4.2/) are used for the quality assessment on the triple collocation analysis.

## 2 Generation of the BEC Arctic SMOS SSS product v3.1

The generation of the improved and higher spatial resolution salinity maps has followed several steps which are described below. A scheme of the whole algorithm is shown in figure 1 (for more details, the reader is referred to the ATBD of the Arctic+ project (Martínez *et al.*, 2020)).

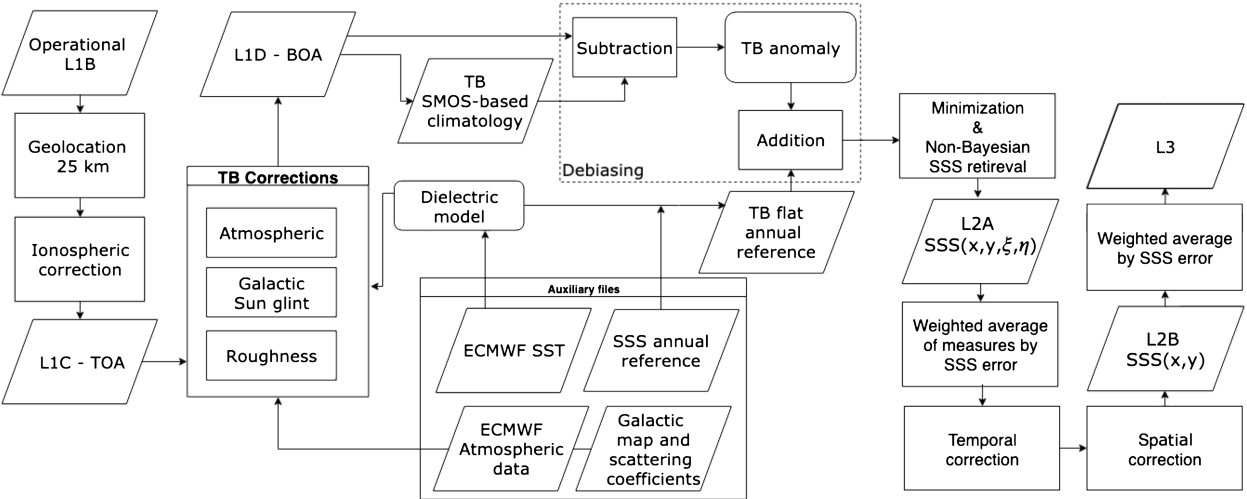

**Figure 1.** General block diagram for Arctic SSS retrieval. The debiased non-Bayesian method was applied as part of the algorithm, but debiasing was done in $TB$ rather than in SSS.



## 2.1 Geolocation and projection of the Brightness Temperatures

We use the ESA Earth Explorer Mission CFI propagation libraries v 3.7.4 ESA (2014) to geolocalize the SMOS $TB$ measurements. The geographic coordinates (longitude and latitude) are transformed to plane coordinates by means of the Lambert Azimuthal Equal Area map projection (LAEA) (Snyder, 1987).

The antenna hexagonal grid in which the $TB$ image is reconstructed contains $64 \times 64$ points (instead of 128 used in the standard SMOS L1 processor). This resolution in the antenna level results in 4096 grid points and allows us to reduce the

computational time without loss of information/resolution. The areas of the Field of View (FOV) containing aliases between different regions of the Earth are discarded. Once all the geolocation magnitudes have been computed and the measured $T_B(\xi_i, \eta_i)$ are known in all $64 \times 64$ FOV points, the Earth grid is generated and the points of this grid are retroprojected up to SMOS antenna coordinate reference system for each SMOS snapshot. In this v3.1 product, we have computed the TBs directly in the final grid of the salinity map products. This choice has been made to keep the maximum information of the salinity

gradients without losing resolution due to interpolation errors.

## 2.2 Computation of the Brightness Temperatures at the ocean surface

The $TB$ transformation from Top of the Atmosphere (TOA) (see the previous section) to Bottom of the atmosphere (BOA) is similar to the one performed in the operational SMOS level 2 processor chain (more details are described in ICM-CSIC *et al.* (2016)). The measured $TB$ by SMOS is the result of different contributions Zine *et al.* (2008): sea emission, atmosphere

emission contribution, galactic contribution, and sun glint.

The galactic and sun glint contributions are computed following the models described in Tenerelli *et al.* (2008) and Reul *et al.* (2007), respectively. We use the roughness model developed by BEC ((Guimbard *et al.*, 2012)). The auxiliary data provided by ECMWF is collocated with SMOS measures and used to evaluate the models. Finally, we compute the $TB$ corresponding to the flat sea contribution ($TB_{flat}^H$ and $TB_{flat}^V$) by subtracting the rest of the contributions. The polarizations of the SMOS TBs

are affected by the ionosphere, and they can be corrected following Zine *et al.* (2008). Nevertheless, the ionosphere produces a rotation between the $TB$ polarizations leaving unaltered the first Stokes parameter ($I = TB^x + TB^y$), parameter used to perform the $TB$ inversion.

## 2.3 Debiasing

The debiased non-Bayesian retrieval method was introduced in Olmedo *et al.* (2017) to retrieve salinity from SMOS $TB$.

Salinity retrieval is performed by means of a minimization of the difference between the measured first Stokes parameter and the modelled one. This minimization follows a non-Bayesian scheme i.e., a SSS value is retrieved for each $TB$ measurement. The new salinity Arctic product also retrieves salinity using a non-Bayesian scheme but introducing important changes.

The SSS debiasing approach employed in the previous version of the BEC Arctic SMOS SSS product (v2.0, (Olmedo *et al.*, 2018)) started from a long-term time series (8 years) of SSS retrievals. These SSS retrievals were grouped according to their

geographical location, incidence angle, distance to the center of the swath (across-track distance), and the satellite overpass





direction. These salinity values were accumulated in classes for each group, obtaining the discrete salinity distribution function for each one. The characteristic value of these distributions can be considered as the SMOS-based salinity climatology. The mean around the mode of each distribution was chosen as its characteristic value. To mitigate the local time-independent biases, the corresponding SMOS climatology is subtracted from each measured SSS value and replaced by an annual SSS reference.

World Atlas Ocean 2013 (WOA) (Zweng *et al.*, 2013) was the reference of choice for v2.0. This method mitigates biases like those caused by land-sea contamination or permanent Radio Frequency Interferences (RFI) sources.

In this work, we aim at improving the algorithm in two specific points. First, the non-homogeneous division of the antenna FOV into groups of incidence angles and across-track distance derives into different statistic representativeness for different points of the antenna, providing a non-optimal resolution of the final climatological salinity product. This has been improved by

introducing an homogeneous discretization of the Extended Alias Free Field of View (EAF-FOV) in $\xi\eta$ coordinates. Secondly, the non-linearity of the L-band dielectric models at very low salinity ranges amplifies the errors of the retrievals at low salinity values. Since the retrieval procedure propagates systematic errors from $TB$ value to the resulting SSS, the debiasing is applied at $TB$ level and not at SSS level, so to mitigate as much as possible these effects (more details can be found in Martínez *et al.* (2020)).

The interferometric nature of SMOS divides the SMOS antenna FOV into a hexagonal grid. Accordingly, the antenna FOV has been homogeneously divided into hexagonal cells that cover the same antenna area. To ensure that a number of measurements large enough are accumulated, by each hexagonal cell, to compute a reliable SMOS-based climatology, each cell contains 7 points of the original $64 \times 64$ FOV grid.

To perform the spatial bias correction, the measured first Stokes parameter is grouped into discrete distributions according to

the antenna cell in which it has been acquired, its geographical location on the Earth, and the satellite flight direction. Thereby, the SMOS-based climatology is subtracted from the individual measures of the first Stokes parameter, and the annual reference is added to it. No brightness temperature exists from World Ocean Atlas, so it is necessary to compute it starting from WOA 2018 SSS and SST (Zweng *et al.*, 2018) using the Meissner and Wentz dielectric model.

### 2.3.1 SMOS-based climatology

The SMOS-based climatology has been computed using the SMOS $TB$ data from the 2013-2019 period (both included). We discarded the years 2011 and 2012 because of the strong affectation of RFI in the earlier years of the mission.

The SMOS-based climatology is performed by computing the distribution of the first stokes separately for ascending and descending orbits. The histograms are created by accumulating valid measures in bins of 1 K for each 25 km EASE-Grid 2.0 North grid point ($x$ and $y$ coordinates) and FOV coordinates ($\xi$ and $\eta$ coordinates). Only latitudes beyond 50°N are considered.

We apply the following filtering criteria over $I$ before computing the climatology:

– Only measurements in the range 75 K $< I_{flat}^{meas} <$ 165 K are considered

– Measurements acquired with ECMWF SIC values larger than 0.05 are discarded.





- The Tukey (1977) rule is used to detect outliers. Then, measures $\Delta$ accomplishing one of these conditions:

$$\Delta > Q_3 + 1.5 \times IQR; \quad \Delta < Q_1 - 1.5 \times IQR$$

are considered outliers. Here $IQR = Q_3 - Q_1$ and $Q_1$ and $Q_3$ are the first and third quartile respectively. Outliers detection is implemented in two stages:

- We compute the linear regression from all the $T_B(\theta)$ measures for a given geographical point obtained at different incidence angles ($\theta$) in a given orbit. Outliers are detected by computing the difference between this linear regression and each individual measurement ($\Delta$).
- The rule is also applied for each $I_{flat}^{meas}(x, y, \xi, \eta)$ distribution.

Once the valid measurements are selected, for each acquisition condition $\gamma = (x, y, \xi, \eta, d)$, with $d$ the orbit direction, we accumulate the $TB$ measurements ($I_{flat} = (TB_{flat}^{H} + TB_{flat}^{V})/2$) acquired under $\gamma$ conditions in $I_{flat}(\gamma)$. The histograms are created adding only valid measurements in bins of 1 K for each 25 km EASE-Grid 2.0 North grid point ($x$ and $y$ coordinates) including the measures collected in a 75×75 km square. The measurements are grouped for each FOV cell composed of seven unshared $\xi$ and $\eta$ coordinates. Only latitudes above 50°N are considered. Then, for each $\gamma$, we compute the following statistical parameters: frequency, mean, median, interquartile range, and the second, third, and fourth central moments. The representative value used afterwards for the debiasing, namely the SMOS-based climatology $I_{\gamma}^{c}$ is assumed, as in the previous version of the debiasing method, as the mean around one standard deviation from the mode of the $I_{flat}(\gamma)$.

## 2.4 Inversion

Once the systematic errors of the measured flat sea $I_{flat}^{meas}$ are corrected, the SSS retrieval can be performed by using a dielectric constant model. The flat sea emissivity is described by Fresnel reflection law that is a function of the incidence angle of the radiation $\theta$ and the dielectric coefficient $\varepsilon$, which depends on the SST, the frequency, and the conductivity, which in turn depends on the salinity.

In the L-band range, the more common dielectric constant models used are Klein and Swift model (KS) (Klein & Swift, 1977) and Meissner and Wentz model (MW) (Meissner & Wentz, 2004; Meissner *et al.*, 2018). These dielectric models are based on the Debye conductivity equation Debye (1970). The MW model interpolates the dielectric constant as a function of salinity between 0 and 40 psu and provides accurate values for the ocean surface temperature between $-2°C$ and $29°C$, while KS is not so accurate for low ocean temperatures (Zhou *et al.*, 2017; Dinnat *et al.*, 2019). Therefore, we have used the MW model to derive the high latitudes SSS.

We obtain the salinity value by minimizing the following cost function:

$$F = \|I_{flat}^{mod} - I_{flat}^{meas}\|^2. \tag{1}$$

with $I_{flat}^{mod}$ the first Stokes described by the models. We use the Newton-Raphson method (Press *et al.*, 1992) to find the SSS value that minimizes the equation $F$.

Earth System Discussions
Science
Data
### 2.4.1 $TB$ filtering approach

We apply the following filtering criteria before the inversion:

- Values of $I^{meas}_{flat}$ too close to the belts and suspenders (closer than 0.025 antenna units) are discarded (see figure 2).

- Points affected by Sun tails or reflected Sun circle are also discarded.

- The $I^{meas}_{flat}$ values considered as outliers during the SMOS-based climatology computation (section 2.3.1) are discarded as well.

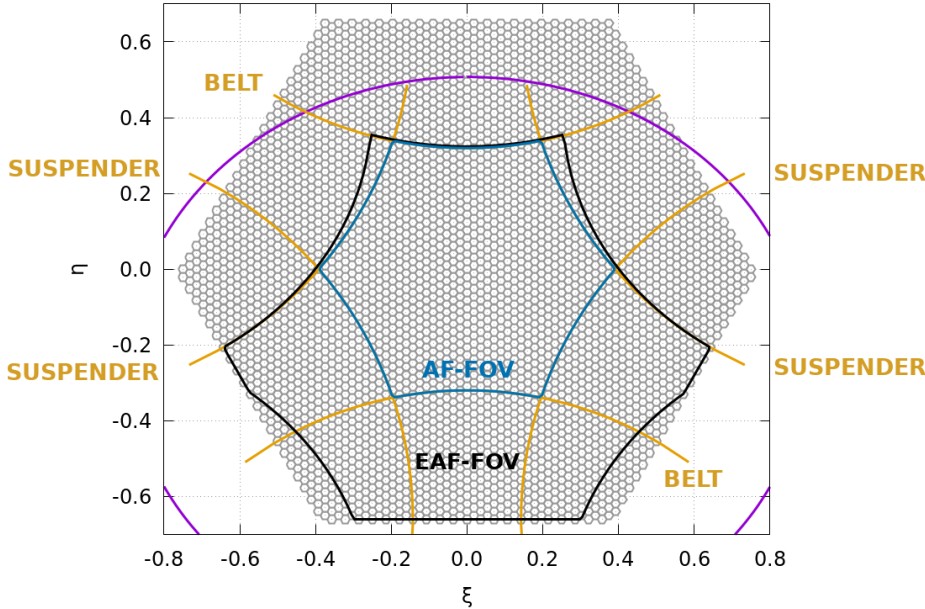

**Figure 2.** $64 \times 64$ SMOS hexagonal field of view. Purple line indicate the Earth limit; beyond this limit the FOV points contain sky $TB$. Blue and black lines encircle the AF-FOV and EAF-FOV respectively. Points in the horizontal yellow lines are known as *belt* whereas *suspenders* are the points included in the vertical yellow lines. Belt and suspenders are points of transition between free-alias zone and zones affected by Earth-sky aliases or between zones affected by different Earth-sky alias, therefore the measures are expected to be somewhat degraded over there.

Additionally, $TB$ values are discarded if the SMOS-based climatology used for its bias correction is considered as a moderately non-normal distribution (kurtosis and skewness conditions according to West *et al.* (1995)), if climatology has been
computed using a low number of measures or if the standard deviation of the climatological distribution is too high close to the coast (suspicion of residual land-sea or persistent RFI contamination). These conditions are summarized in the following rules:



- The minimum number of measures to create the SMOS-based climatology must be 100.

- The absolute value of kurtosis must be less than 7.

- The absolute value of skewness must be less than 2.

- For points located less than 100 km from the coast, only those having a standard deviation less than 8 K are taken into account.

To ensure the minimum ice-sea contamination all points having SIC>0 according to Sea Ice Climate Change Initiative product OSI-450 and OSI-430-b (Lavergne *et al.*, 2019) are not included in the minimization process. The distance to the ice
edge (defined by the line SIC=0) is also stored with the same purpose (minimizing the ice-sea contamination by avoiding the points too close to the ice in the L3 map generation).

### 2.4.2 Minimization and error propagation

The $I_{flat}^{meas}$ accepted after the filtering process are debiased by extracting the SMOS climatologic value and by adding the WOA derived climatological value and introduced in the minimization function. We apply the following convergence criteria in the
iterative scheme:

- The change in salinity values between two consecutive iterations is less than 0.001.

- The percentage of variation in the cost function between consecutive steps is less than 1.

- The above two conditions are accomplished during 5 consecutive iterations to avoid oscillatory solutions.

- The above condition is accomplished in less than 150 iterations.

The propagation of the $TB$ radiometric error to salinity is made by performing the minimization of two additional quantities:

$$
\begin{aligned}
I_{flat}^{meas-}(x,y,\xi,\eta,d) &= \frac{1}{2}(TB_{flat}^H(x,y,\xi,\eta,d) - \sigma^H + TB_{flat}^V(x,y,\xi,\eta,d) - \sigma^V) \\
I_{flat}^{meas+}(x,y,\xi,\eta,d) &= \frac{1}{2}(TB_{flat}^H(x,y,\xi,\eta,d) + \sigma^H + TB_{flat}^V(x,y,\xi,\eta,d) + \sigma^V)
\end{aligned}
\tag{2}
$$

here $\sigma^{[H,V]}$ is the radiometric accuracy of $TB_{flat}^{[H,V]}$. Then, the salinity error is computed from the following equation:

$$
\epsilon(x,y,\xi,\eta,d) = \frac{1}{2}|SSS(x,y,\xi,\eta,d)^{meas+} - SSS(x,y,\xi,\eta,d)^{meas-}|
\tag{3}
$$

where $SSS^{meas+}$ and $SSS^{meas-}$ are obtained by the inversion of $I_{flat}^{meas+}$ and $I_{flat}^{meas-}$, respectively.





### 2.4.3 SSS filtering approach

Once each orbit has been processed, we perform the inversion using the mentioned MW dielectric model to create orbits that contain one salinity value for each measured $TB$. At this level, we obtain for each location as many salinity values as measurements have been inverted. Thus, the salinity is a function of the incidence angle. To create this product we apply the following filtering criteria:

- $SSS(x,y,\xi,\eta,d) < 0$ and $SSS(x,y,\xi,\eta,d) > 50$ are discarded.

- In the case of $SSS(x,y,\xi,\eta,d) > 25$, we discard the retrieval when $|SSS(x,y,\xi,\eta,d) - SSS_{woa2018}| > 7$.

- In the case of $SSS(x,y,\xi,\eta,d) < 25$, we discard the retrieval when $|SSS(x,y,\xi,\eta,d) - SSS_{woa2018}| > 21$. We relax the criteria in this case to better capture the river discharges and melting episodes which are not well described in WOA 2018.

## 2.5 Generation of SSS satellite overpasses

Prior to the L3 maps generation, we create the L2B product. The L2B orbits provide salinity values independent from the antenna point acquisition and are generated from L2A snapshots by weighted averaging all the measures obtained for a given grid point. Outliers detection is performed in a similar way as it was applied to TB: a linear regression is performed from all the incidence angle sorted SSS values of the same orbit obtained for a given geographical point. The outliers are detected, according to Tukey (1977) rule, from the difference between this linear regression and each individual measure. L2B SSS values are only computed for those grid points containing more than 12 unfiltered L2A retrieved SSS values.

Assuming the weight function as the inverse of the squared error of each L2A SSS measure, we ensure that the measures coming from $TB$ having a high radiometric error will have a small influence in the obtained value for SSS at L2B level. Therefore, the average of all of the SSS retrievals for a given location is weighted in the following approach:

$$SSS(x,y,d) = \frac{\sum_{\xi,\eta} SSS(x,y,\xi,\eta,d) w(x,y,\xi,\eta,d)}{\sum_{\xi,\eta} w(x,y,\xi,\eta,d)} \qquad (4)$$

where the weight function is given by

$$w(x,y,\xi,\eta,d) = \frac{1}{\epsilon^2(x,y,\xi,\eta,d)}. \qquad (5)$$

where $\epsilon(x,y,\xi,\eta,d)$ is given by expression 3. Therefore, the error of each L2B salinity value is given by the expression

$$\epsilon(x,y,d) = \frac{1}{\sqrt{\sum_{\xi,\eta} 1/\epsilon^2(x,y,\xi,\eta,d)}}. \qquad (6)$$

## 2.6 Temporal Bias correction

The $TB$ debiasing procedure only accounts for the spatial biases, so that a temporal correction is also required. The correction implemented in the previous version operates over L3 maps and is based on Argo profiles. The median of the differences





between the collocated L3 SSS and the Argo available for the 9-day period of each map is removed to the corresponding 9-day SSS map. Since we also need to perform a time correction on L2 SSS orbits, and the amount of ARGO data available for
each orbit is insufficient, we propose to perform the temporal bias corrections using the the Global Ocean Forecasting System (GOFS) 3.1 (HYCOM + NCODA) as reference. The correction is based on the iterative scheme presented in figure 3 and detailed as follow:

- Step 1: We start subtracting 12 psu to all SSS (gridpoint level) retrievals. This intends to reduce the number of iterations in the loop and to improve the convergence. The value of 12 psu has been obtained as the better one after several retrieval
processings.

- Step 2: We apply the filtering criteria described in section 2.4.3 and we average the corresponding filtered SSS retrievals in each grid point to generate the satellite overpass (as described in section 2.5)

- Step 3: We update the temporal correction value with the mean difference between each value of the SSS (at orbital level) and the collocated HYCOM salinity.

- Step 4: We add this temporal correction to each $SSS(x, y, \xi, \eta, d)$ at snapshot level.

- We repeat Step 2-4 until the difference between two consecutive corrections is lower than 0.01 psu.

Only orbits providing at least 50 common grid points with HYCOM are considered. Due to the fact that HYCOM provides too salty values in the river mouths, only grid points having a retrieved salinity value above 25 psu and an error below 2.5 psu are considered to compute the temporal correction.

**2.7   Correction of the residual spatial bias**

As it has been described in section 2.3, the debiasing method is based on the substitution of the SMOS-based $TB$ climatology by the SSS reference from the WOA 2018 (Zweng *et al.*, 2018). With that method, a first-order spatial correction is performed.

After the debiasing method, the average salinity obtained for the period used to compute the SMOS-based climatology (years 2013-2019) should have a spatial distribution very close to the reference used to carry out the debiasing (WOA2018). However,
the difference between the weighted averaged of all SSS orbits from years 2013 to 2019 and the averaged SSS from WOA is not close to zero, and significant differences are observed (not shown).

The cause of the differences between ascending and descending passes is mainly linked to the different performances of the ascending and descending debiasing. Depending on the coast orientation, a given point can be affected by land-sea contamination in ascending or descending passes differently, hence, requiring a different correction, too. A similar behaviour is observed
with ice-sea contamination and when RFI are present.

The first Stokes distributions provided by SMOS have generally positive skewness. This means that its representative (the mean around the mode) at each geographical point generally does not match with the mean of the distribution. On the other hand, the WOA 2018 salinity is obtained through an objective analysis scheme using a correction factor given by a weighted

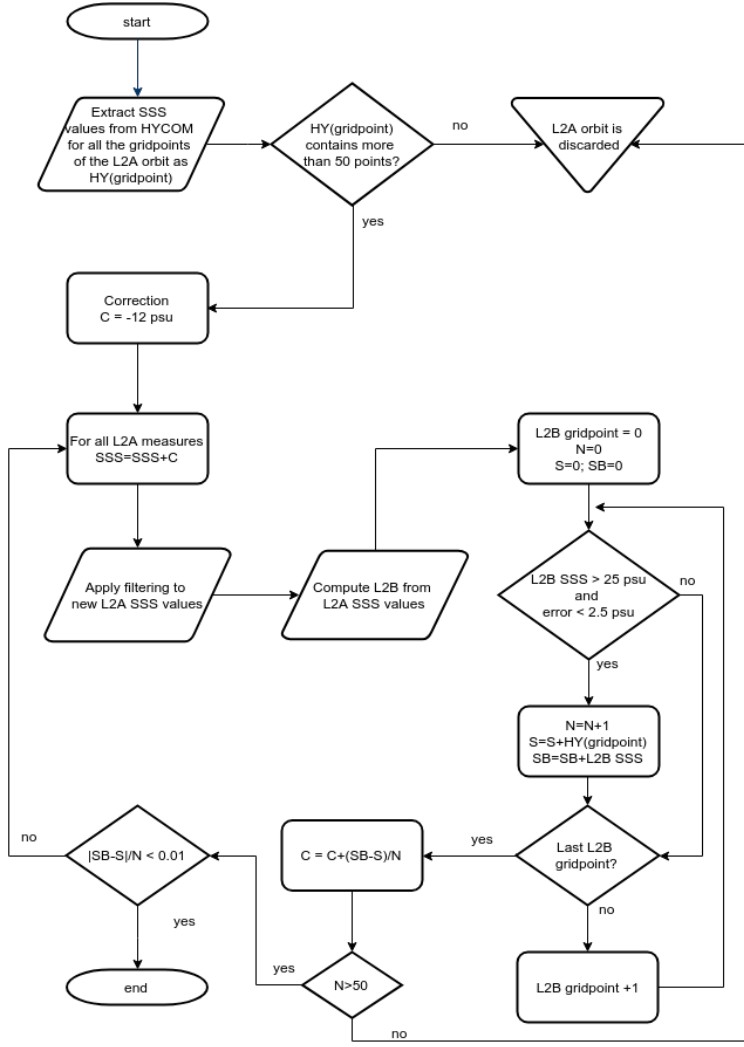

**Figure 3.** Scheme of the iterative procedure used to correct the temporal salinity bias on level 2.

average of the in situ measurements in a given limited region (Zweng *et al.*, 2018, sec. 3.2) assuming, therefore, Gaussianity in

this region. The substitution of the SMOS-based climatology by the $TB$ reference obtained from WOA 2018 salinity introduces

inaccuracies due to the skewness and other second order statistical properties of the SMOS measurements.

In order to mitigate this residual spatial bias, we compute an anomaly spatial map by applying the following algorithm to all the L2B orbits in the period 2013-2019 (shown in figure 4):

– We compute the difference between SMOS overpasses salinities and the ones provided by WOA 2018.

290        –  We generate two anomaly files: one for ascending and one for descending passes (they show differences of up to +/- 1.2
           psu).

After the creation of these two anomaly maps, every L2B salinity value is corrected by subtracting this spatial anomaly. The
salinity values corresponding to geographical points where the anomaly was computed using less than 50 collocations are
discarded.

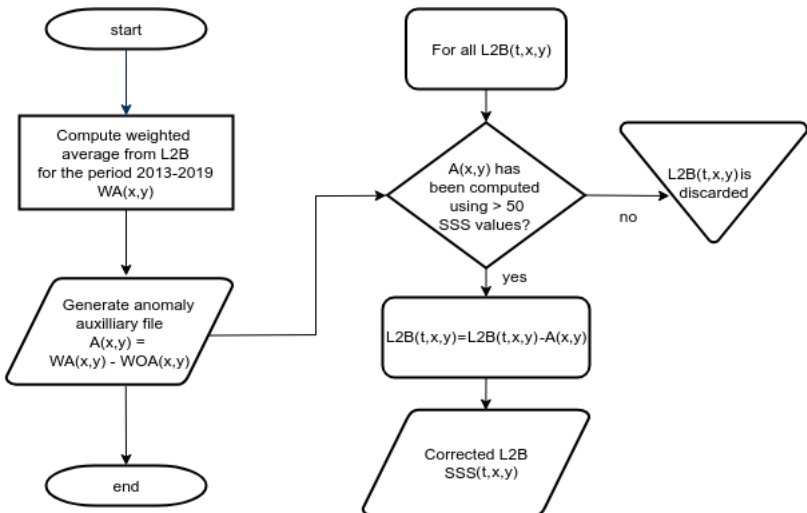

**Figure 4.** Scheme of the procedure used to perform the residual spatial correction on level 2.

**2.8   Generation of L3 SSS maps**

L3 maps are daily generated for 9-days periods. Each map is obtained by a weighted average of all the SSS in all the overpasses
of the 9-day period. Each L2B salinity value is weighted according to its salinity error as described in section 2.4.2. To
minimize ice-sea contamination and land-sea contamination, the L2B points closer than 35 km to the ice edge or coastline are
not considered during the process of L3 maps creation.

300        Figures 5 and 6 show an example of the resulting 9-days L3 salinity maps (fig. 5 a) and the salinity error (fig. 5 b) derived
from the radiometric uncertainty. It should be noticed the greater coverage and detail of the gradients of Arctic+ v3.1 product
to that obtained from the previous BEC Arctic v2.0 product (fig. 5 a-c and 6).

**3   Quality Assessment**

The validation of the product is done using different in situ measurements, the results are described in detail in the Product Vali-
dation Report (Arias *et al.*, 2020) that can be found on the project web page (https://arcticsalinity.argans.co.uk/documentation/).
           We recall some important aspects of this product and its validation procedure:



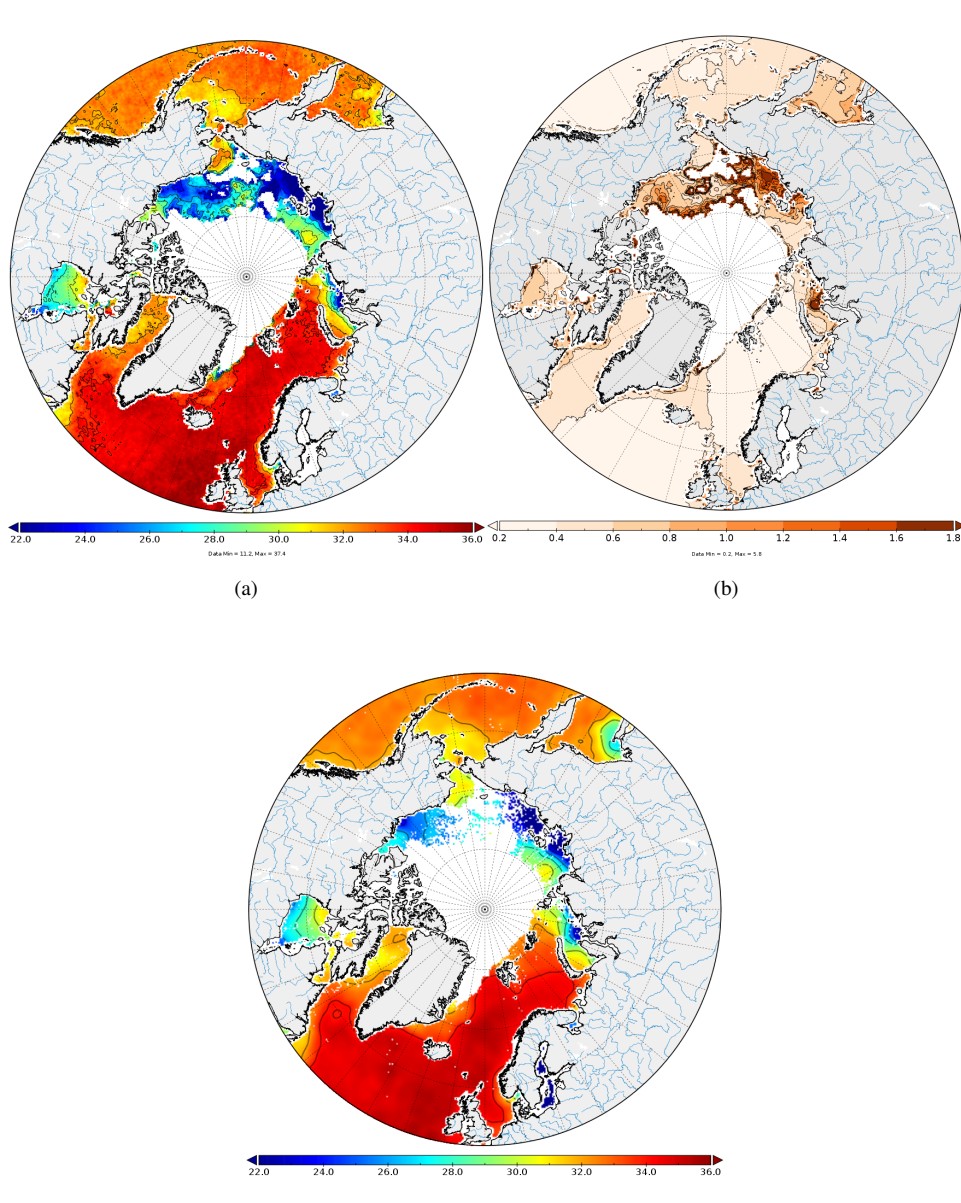

**Figure 5.** (a): Arctic+ v3.1 9-day map of the period August 11-19, 2012. (b): Salinity error derived from the radiometric error. (c): Arctic+ v2.0 9-day map of the period August 11-19, 2012. A section of this map from the Barents Sea to the East Siberian Sea is shown in figure 6

- The validation in the Arctic region is rather complex due to the heterogeneity of the in situ datasets, with a lack of temporal or spatial synopticity. Some of the sources only represent specific regions, with the risk of inducing bias by spatial selection when assessing the product entirely. Others cover a larger spatial representation, however, a lack of a


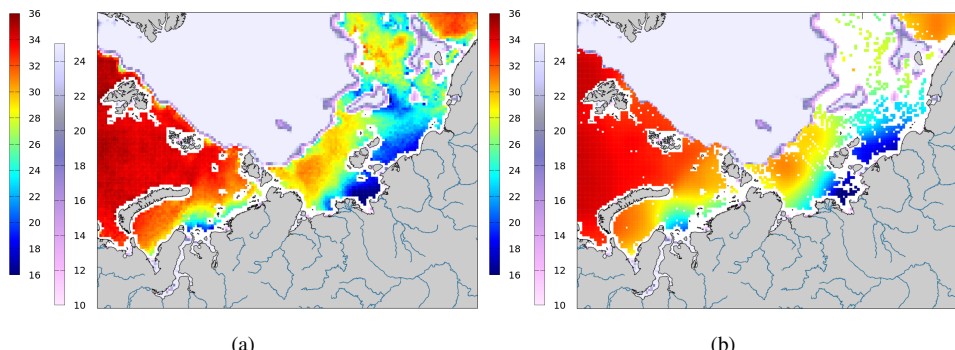

(a)                                                            (b)

**Figure 6.** (a): Detail of the Arctic+ v3.1 product from Figure 5 together with the minimum Sea Ice concentration provided by OSI SAF for the period August 11-19, 2012. (b): Same as (a) but for Arctic+v2.0. The right color bar indicates Sea Ice concentration whereas the left color bar indicates salinity.

proper temporal variability still remains. None of the datasets used can describe both aspects simultaneously, thus the validation requires an exhaustive analysis of the results to assess the quality of the product.

– Arctic+ SSS v3.1 product introduces an improvement in the number of SSS retrievals obtained from SMOS $TB$ for the Arctic region (as shown in Fig. 5 and 6). This represents a significant reduction of data gaps.

– The new product benefits from a polar grid in EASE v2.0 format, which is a standard for the research and operations in the region, improving its usability.

– Arctic SSS+ v3.1 product has been built only using WOA 2018 and HYCOM model output, without using ARGO data.

– The validation of the product using ARGO floats in the Arctic is only valid for Greenland and Norwegian seas regions, where the Argo floats are present. However, a comparison with punctual measurements can not evaluate the improved data coverage neither spatial resolution. Thus, the ARGO analysis can not be used alone to describe the quality of this product.

– It is also important to highlight that, when inter-comparing satellite-based SSS products, there is a need to focus on the selected projections and grid. A fair set of metrics for inter-comparison is only possible over common points resolved at the same spatial scales. This means that the quality control for these products requires setting the products into the same spatial grids and projections. By not doing so, significant errors may be introduced artificially in the metrics, products may be penalized because of differences in the sampling strategies and thus, the match-up databases do not yield the same points and hence, information.



## 3.1 Comparison with ARGO data

Even though the ARGO floats are very scarce in the Arctic and are mainly in the Atlantic and Pacific regions (see figure 7), we
have used Argo SSS data to assess the biases and the standard deviations of the errors of the new SSS products, but with the

330  caution that this analysis is only valid for the region where Argo floats are present, and assuming that Argo values represent a
ground truth for the whole pixel.

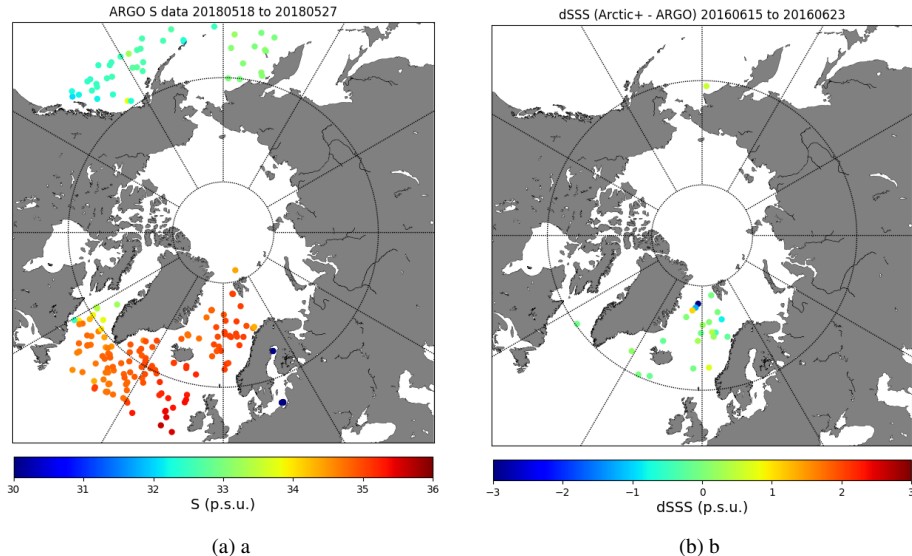

**Figure 7.** (a) Example of total daily ARGO profiles matching the 9-day period associated to one Arctic+ v3.1 product (18/05/2018 to
27/05/2018). (b) Difference between SMOS SSS v3.1 and Argo data after selecting only valid match-ups for product from 15/06/2018.

The methodology followed to perform the temporal and spatial collocation between the Argo SSS with the SSS maps is the
following: For a given in situ point, the closest satellite point is searched both in time and in space, with a radius of 25 km from
the in situ measurement and a maximum period of 9-days off in time. This strategy leads to some repetition in the use of the in

335  situ data points for different maps, but never over the same daily product. This has been deemed as the most solid strategy, as
it maximizes the quantity of in situ information to validate the satellite products.

The statistics (bias and standard deviation) for the differences between Arctic+v3.1 and Arctic+v2.0 with respect Argo
measurements are shown in figure 8 for the period 2011 to 2017. The values for the complete period for v3.1 are: $Mean =
0.02$, $STDD = 0.39$, $RMSD = 0.39$ and correlation $R = 0.94$. The values for v2.0 are: $Mean = -0.01$, $STDD = 0.28$,

340  $RMSD = 0.29$ and correlation $R = 0.97$. The standard deviation is larger than the previous BEC v2.0 product, but this is
expected since the BEC Arctic v2.0 product was generated by performing objective analysis with correlation radii 321 km,
267 km, and 175 km (Olmedo *et al.*, 2016). The large correlation radii produce a large smoothing effect, reducing the noise.



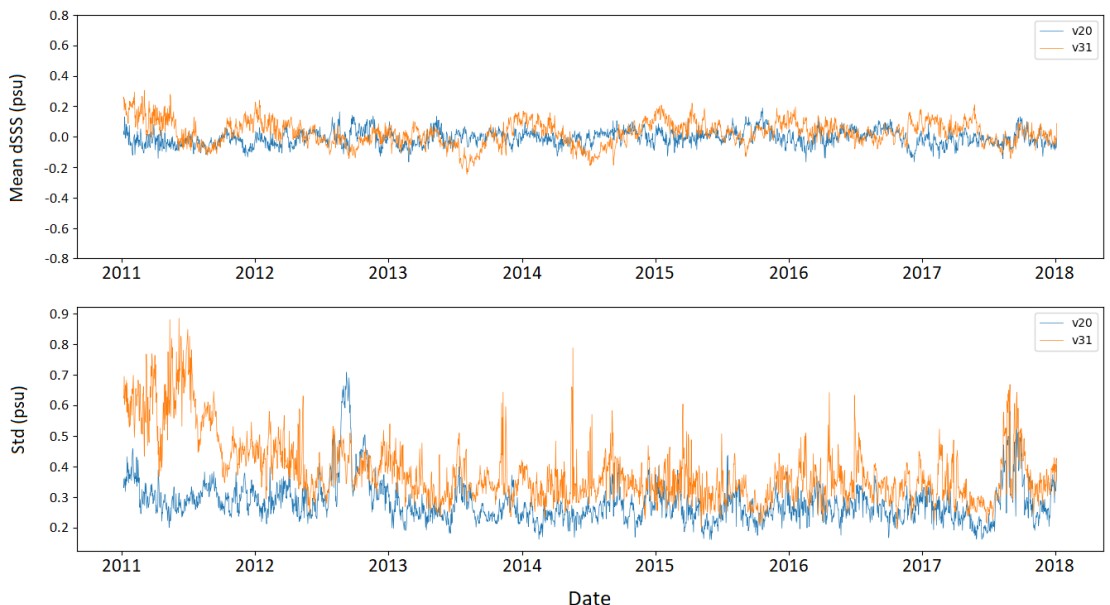

**Figure 8.** Results of time series of metrics for bias and standard deviation per each 9-day map of the satellite-based products. Blue line corresponds to BEC v2.0 and orange line to Arctic+ v3.1.

However, the smoothing results in a reduction of the spatial resolution, which is significantly improved in the v3.1 product (see section 3.3). BEC v3.1 product, thus, contains more dynamic information than v2.0.

The degraded quality of data in the first two years (2011 and 2012) are due to a high RFI source located in Greenland which highly contaminated the measurements and those could not be corrected during the processing step, so we recommend the users to avoid those tow years data. Latter on, by end of 2012, the main source of RFI was locked down, permitting to have higher quality measurements (Oliva *et al.*, 2016).

### 3.2    Comparison with TARA dataset

TARA salinity data presents a large range in the spatial variability of salinity between 26 and 35 psu in the Arctic Ocean (see figure 9.

The mean, the standard deviation (STDD), root mean square (RSMD), and correlation (R) are computed for all the residuals of the collocated points between TARA and the Arctic+ v3.1 (Table 1). To assess the quality of the Arctic+ v3.1 product over a specific region, a splitting of the TARA transect has been applied grouping data into sub-basins: the Norwegian Sea, 355    Barents, Kara Sea, Laptev Sea, East Siberian Sea, Chuckchi Sea, Beaufort Sea, Baffin Bay. The results are shown in the Table 1.

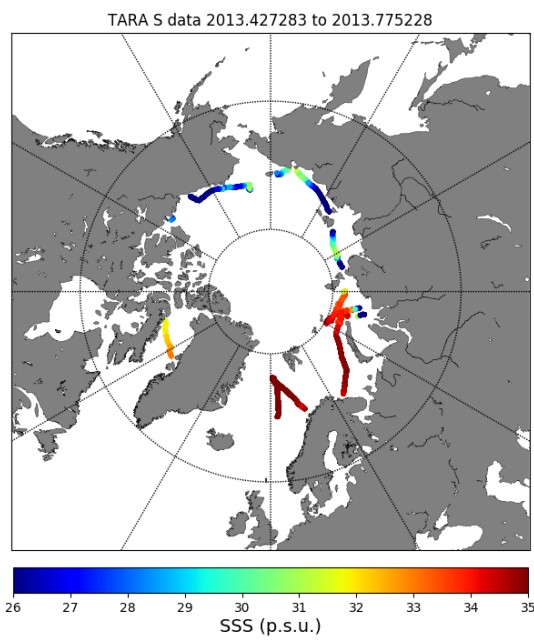

**Figure 9.** TARA expedition TSG measured SSS values. TARA expedition circumnavigated the Arctic between June and October 2013 following a counterclockwise path. SSS data reveals the high spatial variability of SSS in the Arctic, as result of the multiple sources of SSS variability (river tributaries, ice melting).

**Table 1.** Validation results of Arctic+ v3.1 and BEC v2.0 in Arctic Sub-basins using TARA TSG data

|  | full | Norweg. | Barents | Kara | Laptev | East Sib. | Chuckchi | Beaufort | Baffin |
|---|---|---|---|---|---|---|---|---|---|
| Arctic+ v3 |  |  |  |  |  |  |  |  |  |
| Mean | -0.07 | -0.09 | -0.62 | -0.8 | 0.72 | 0.42 | 2.44 | 2.26 | 0.61 |
| STDD | 1.52 | 0.22 | 0.21 | 1.40 | 2.60 | 1.32 | 1.41 | 0.93 | 0.31 |
| RMSD | 1.52 | 0.23 | 0.66 | 1.62 | 2.70 | 1.39 | 2.82 | 2.44 | 0.68 |
| R | 0.91 | 0.52 | 0.48 | 0.89 | 0.80 | 0.94 | 0.40 | 0.98 | 0.62 |
| BEC v2 |  |  |  |  |  |  |  |  |  |
| Mean | -0.54 | -0.06 | -0.49 | -1.78 | 0.48 | -0.74 | 1.34 | 2.74 | 0.03 |
| STDD | 1.63 | 0.16 | 0.31 | 1.32 | 2.44 | 1.22 | 1.50 | 0.207 | 0.18 |
| RMSD | 1.71 | 0.17 | 0.58 | 2.21 | 2.49 | 1.43 | 2.01 | 3.43 | 0.19 |
| R | 0.88 | 0.59 | 0.31 | 0.90 | 0.84 | 0.94 | 0.17 | 0.95 | 0.87 |

Matchups with TARA show good results of the Arctic+ v3.1 product, better than the previous BEC v2.0 product in most of the seas (see Product Validation Report, (Arias *et al.*, 2020)).





Nevertheless, metrics with TARA are not simple to interpret due to the lack of synopticity into the dataset (acquired over a
relatively long period of time, i.e., representing different observational conditions) and the lack of spatial homogeneity of the
sampling, which explains the relatively large variability observed in the metrics for different sub-basins.

## 3.3   Error characterization by Correlated Triple collocation

Triple Collocation (TC) is a method originally introduced by Stoffelen (1998) to provide estimates of the measurement error
variances of three systems measuring the same variable at the same time. TC is based on the statistical relations between the
measurement variances and covariances to deduce the error variances for each measurement. TC requires to have a series long
enough series of collocated triplets of the measurements to obtain reasonable estimates of the second-order moments of those
measurements.

Besides, it is usually required that the three measurement systems are completely independent, with different space-time
acquisition scales, and thus the so-called representativity error must be properly accounted for (Stoffelen, 1998; Hoareau *et al.*,
370   2018).

Recently, a variant of TC especially adapted to deal with remote sensing measurements, has been introduced: The Correlated
Triple Collocation (CTC) (González-Gambau *et al.*, 2020). When applying CTC, the data is assumed to have the same space-
time sampling, that is, they represent the same spatial and time scales. In contrast with standard TC, it is assumed that two of
the datasets can have correlated errors (for instance, they are derived from the same basic measurement system). Besides, and
considering that remote sensing series are typically not too long, CTC is optimized to provide reasonably good estimates of
the error variances even with a limited number of samples. With those conditions, CTC can be used to obtain maps of error
variances of triples of remote sensing SSS maps and obtain, for example, a different map for every year.

We have applied the CTC, following González-Gambau *et al.* (2020), to characterize the SSS errors with 2016 data. We
have taken three sets of collocated SSS maps: JPL SMAP v4.2 SSS, 8-day maps; BEC SMOS Arctic SSS v2.0, 9-day maps
and BEC SMOS Arctic+ v3.1, 9-day maps. We have considered the products reduced to the common resolution (that of BEC
v2.0).

The correlated triple collocation analysis helps to assess properly the differences existing between the derived satellite
products. Figure 10 shows the estimated error standard deviation for each one of the three datasets. The differences between
the products are shown in figure 11. Over the majority of the Arctic, BEC v3.1 has the smallest error, except in some specific
regions where BEC v2.0 is better (Hudson Bay, east coast of Greenland, and Kara Sea). JPL 4.2 is in all cases the product with
the greatest error.

## 3.4   Spectral Analysis

The analysis of spectral slopes permits obtaining information about the effective spatial resolution of the different remote sens-
ing datasets. Theoretical studies have reported that Power Density Spectra (PDS) slopes are expected to range between -1 and
-3, depending on the dynamical regime that drives the ocean (Blumen, 1978; Charney, 1978). Moreover, the presence of white
noise makes the spectral slope tends to 0 (log PDS vs log wavenumber bend and become horizontal at high wavenumbers). In

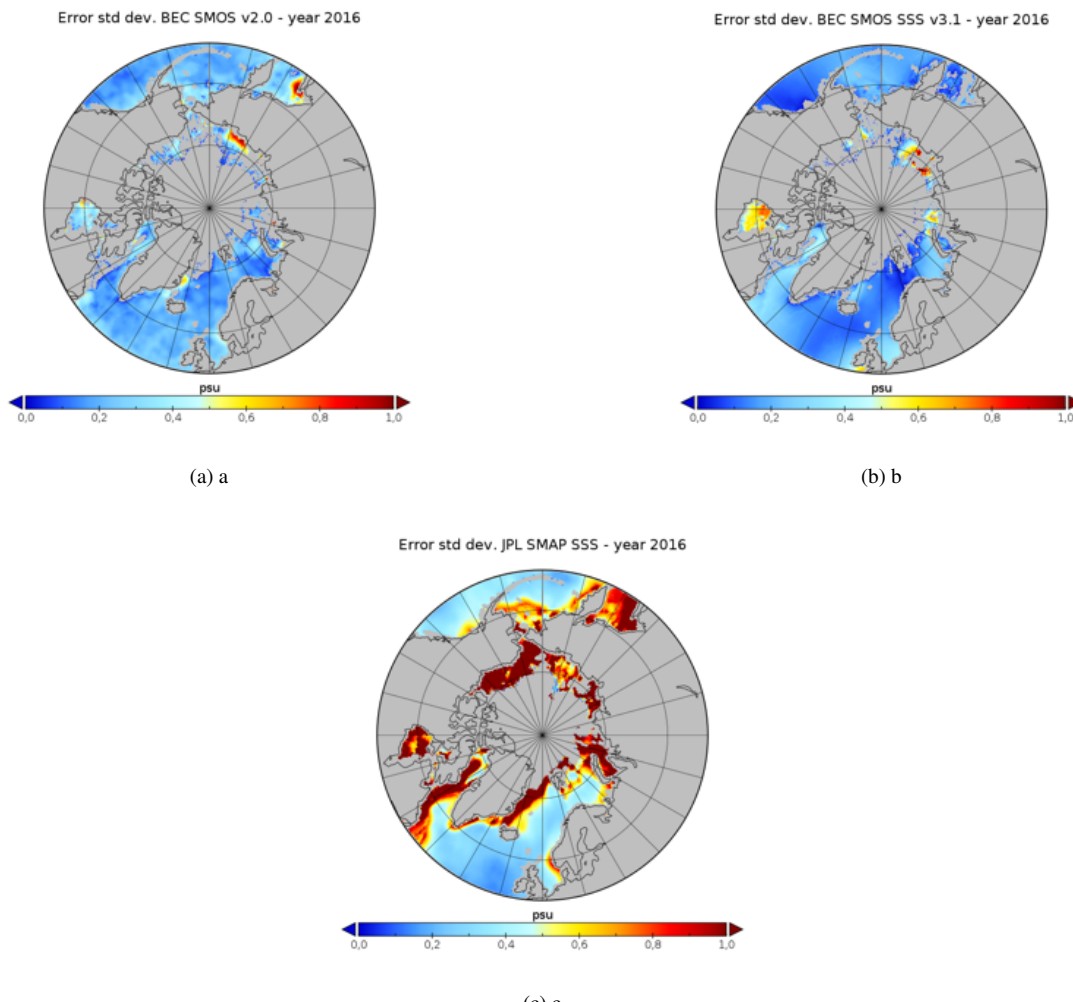

**Figure 10.** Error standard deviations computed via Correlated Triple Collocation for BEC SMOS Arctic SSS v2.0 (a), BEC SMOS Arctic SSS v3.1 (b) and JPL SMAP SSS v4.2 (c), for all the collocated maps in the year 2016.

contrast, when the spatial resolution of the data is over smoothed, a systematic lack of energy appears at high wavenumbers and a faster decay is observed on the spectral slope for the wavenumber larger than the effective resolution threshold. In this analysis, we use the value of the spectral slope k-2 (Blumen, 1978; Charney, 1978) as reference.

The spectral analysis approach has been applied as in Hoareau et al. (2018), over three regions:

- – Bering Strait (155°E-130°W & 70°-72°N)

- – Laptev Sea (115°-170°W & 76°-78°N)

- – Nordic Seas (4°E-5°W & 63°-80°N)



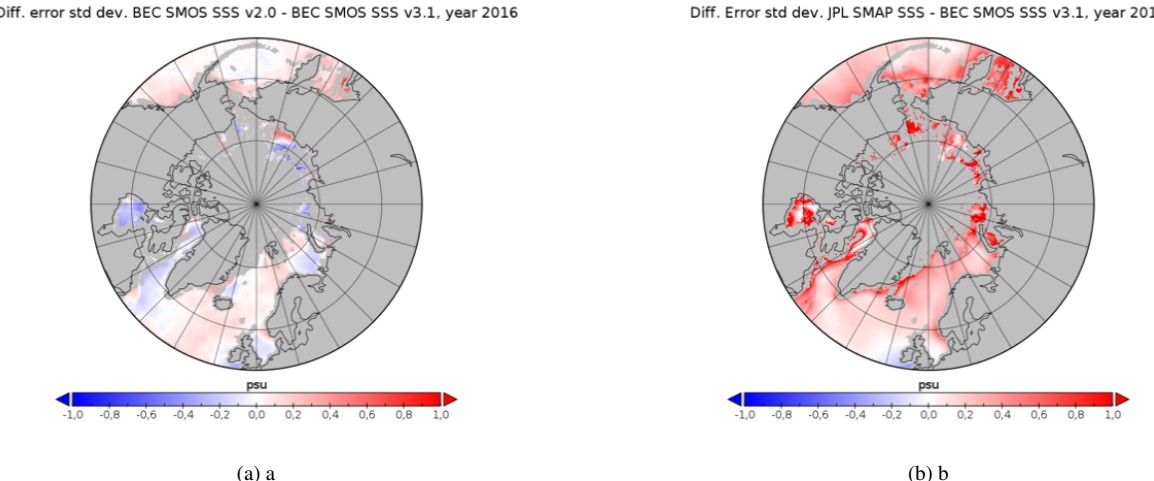

**Figure 11.** Difference between the error standard deviations of BEC SMOS SSS v2.0 (a) and of JPL SMAP SSS v4.2 (b) with BEC SMOS SSS v3.1 for the year 2016.

For each region and product, we compute the PDS over each 9-day map and then average the PDS over the full year 2016
(figure 12)). Notice that we also compute the average of the PDS for the summer period (months from June to October), when sea ice coverage is lower (figure 13), to reduce the fluctuations of each individual spectrum. PDS are given as a function of wavenumber values in degrees (latitude degrees for meridional regions, i.e., the Nordic Sea, and longitude degrees for zonal regions, i.e., the Laptev Sea and Bering Strait) and as wavelength values in kilometres.

First of all, note that the PDS shapes of the data in all regions are similar when averaging the spectra over all the year (figure
12) or only over the ice-free months, i.e., from June to October (figure 13).

The level of noise for each remotely sensed product produces small fluctuations in the shapes of the PDS. Despite this, they follow a slope of -2 indicating that the geophysical structures of the SSS data are consistent until a 50 km wavelength for the case of Arctic+ SSS v3.1 (blue line) and for SMAP JPL (red line) in all regions. This wavelength corresponds to a spatial resolution of 25 km.

In contrast, the BEC SSS v2.0 PDS (magenta line) is able to consistently describe the geophysical structures up to 250 km wavelength (PDS slope similar to -2). For smaller scales there is a faster decay of the PDS slope, indicating a loss of signal, specially in the Nordic Seas and the Bering Strait, probably due to an over smoothing in the optimal interpolation algorithm.

Therefore, the Arctic+ v3.1 data is the one containing the most consistent spatial representation at smaller scales, allowing a more accurate description of Arctic SSS processes.



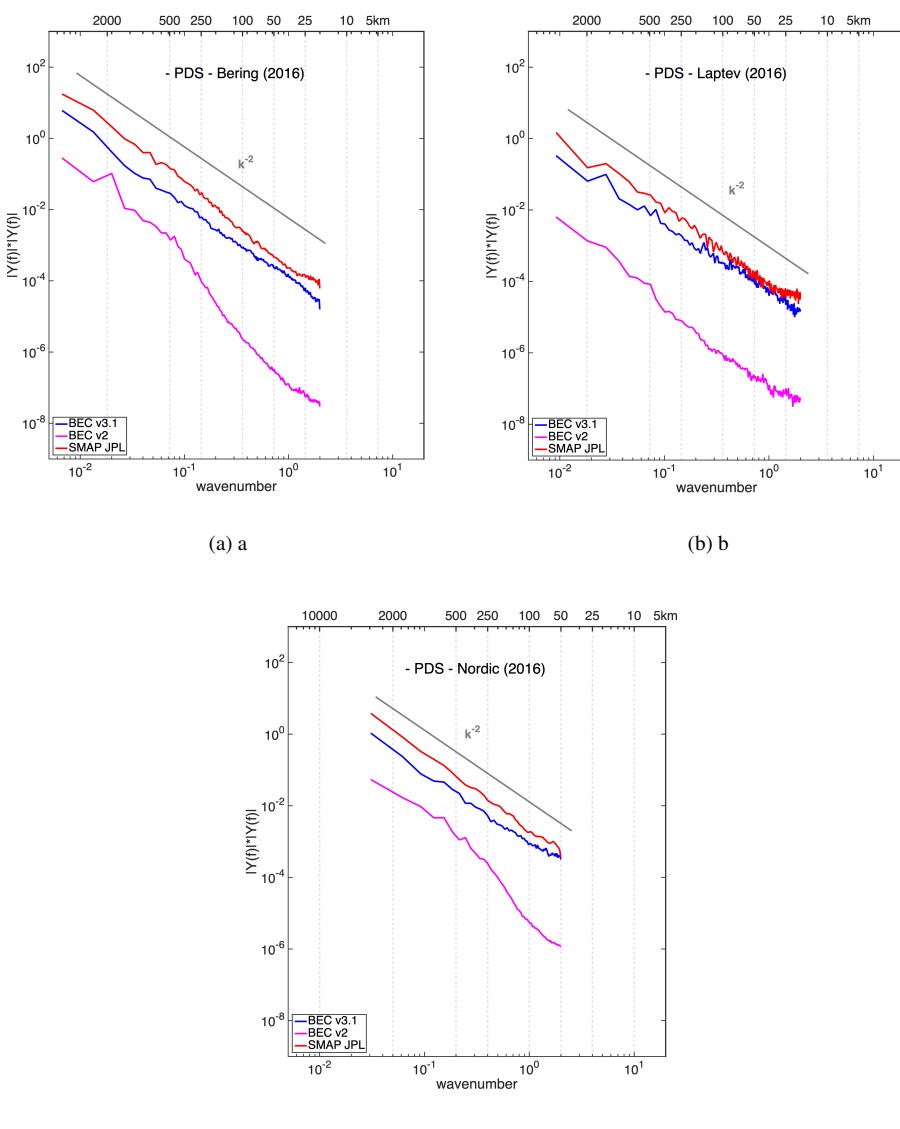

**Figure 12.** Spectral analysis for Arctic+ v3.1 , BEC Arctic v2.0 and SMAP products during the whole 2016 for different regions Bering Strait (a) Laptev Sea (b) and Nordic Sea (c).

## 4 Conclusions

This paper presents the methodologies used to produce the new enhanced Arctic+ SMOS SSS v3.1 product developed under the context of the ESA Arctic+Salinity project (AO/1-9158/18/I-BG).



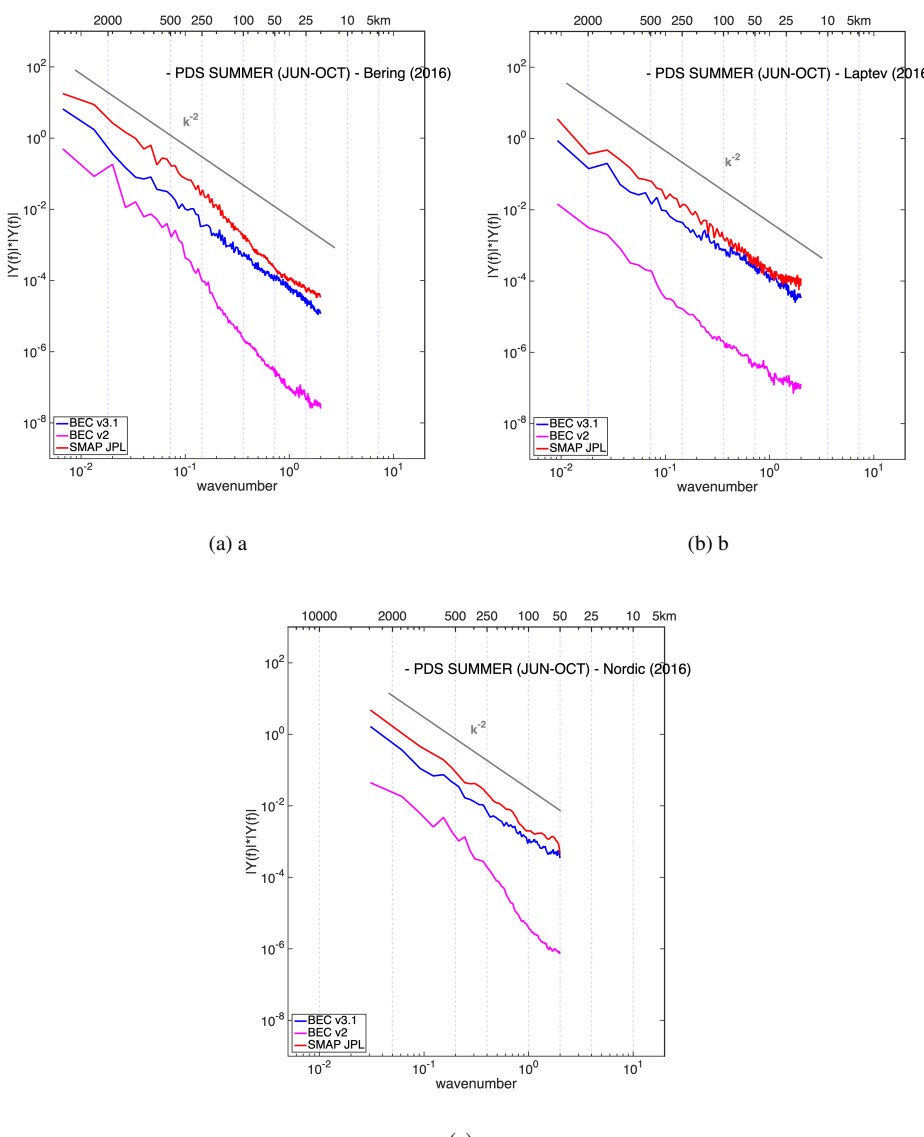

**Figure 13.** Spectral analysis for Arctic+ V3.1 , Arctic+v2 and SMAP products for summer (June to October) 2016 for different regions Bering Strait (a) Laptev Sea (b) and Nordic sea (c).

The inversion is performed by using the debiasing non-bayesian method, as described in Olmedo *et al.* (2018) and Olmedo *et al.* (2021), but performing the debiasing at $TB$ level (refer Martínez *et al.* (2020) for more details) while in BEC v2.0 the debiasing is performed at SSS level. Moreover, the temporal bias correction is performed here using GOFS 3.1 (HYCOM + CONDA) data, differently from the BEC v2.0 method which uses the ARGO data.

The SSS maps are produced here by averaging only in time (9 days), but not in space (keeping the same $TB$ resolution). Therefore, there is no loss of effective spatial resolution as compared to $TB$. This finer spatial resolution is one of the main advantages of this product, as shown by the spatial spectral analysis (section 3.4). This new product is preferable to perform studies of the Arctic ocean SSS processes and dynamics.

Arctic+ SSS v3.1 product spans from 2011 to 2019 and consists in daily maps of 9-day averaged, in an equal-area grid at 25km (EASE 2.0 grid). Maps of 3-days and 18-days for the same period and grid are also served in the webpage, but those products have not specific validation results. Furthermore, the swath SSS product (L2) used to generate the maps is also available at the BEC webpage.

The conclusions of the validation procedure are summarized in the following points:

– Arctic+ v3.1 has, in general, a better skill to describe horizontal SSS gradients than BEC v2.0, with better effective spatial resolution. This comes, however, at the price of an increase of bias and a larger uncertainty in some regions.

– Comparison with TARA datasets, Arctic+ v3.1 show good agreement with in situ data for some key areas, like the Beaufort Sea, which is one of those areas in the focus of the Arctic scientific community.

– Comparison with ARGO show that v3.1 has slightly larger RMSD, but present higher correlation with in situ data. It has been stated that the high spatial resolution of v3.1 produce this larger RMSD when comparing with punctual measurements.

– The introduction of the correlated triple collocation also helps to assess properly the differences existing between the current (in 2021) derived satellite products. The metrics show that Arctic+ v3.1 dataset is the best of the three products used in the collocation exercise (BEC v2.0, Arctic+ v3.1, and SMAP JPL v4.2). In particular, the triple collocation shows that of the three, SMAP data yields the largest errors.

– The outcome of the correlated triple collocation is interesting because this technique mainly addresses random error levels, rather than systematic errors. As the method extracts the expected natural variability from the common information between the compared products, it means that the fraction of information in the Arctic+ v3.1 product is the largest of the three products. Mind that triple collocation using these products is relative, as the lack of an in situ source within the set of datasets used in this metric implies that ground truth information is obtained by pure triangulation of the three observables.

– The results of the spatial spectral analysis confirm that Arctic+ v3.1 data is the one containing the most consistent spatial representation at smaller scales than SMAP and BEC v2.0, allowing a more accurate description of Arctic SSS processes.





## 5  Data availability

The product (Martínez *et al.*, 2020) is freely distributed in the BEC (Barcelona Expert Center) webpage (http://bec.icm.csic.es/) with the DOI number: 10.20350/digitalCSIC/12620 and in Digital CSIC server https://digital.csic.es/handle/10261/219679. Data can be downloaded from the ftp service http://bec.icm.csic.es/bec-ftp-service/.

The maps are distributed in the standard grid EASE-Grid 2.0, which has a spatial resolution of 25 Km. In addition to the product validated in this work (L3 with temporal resolution of 9 days), L3 products having a temporal resolution of 3, and 18 days and L2 product are available. These Arctic SSS products cover the period from 2011 to 2019.

*Author contributions.*  J. Martínez, C. Gabarró, A. Turiel, E. Olmedo, V. González-Gambau performed the algorithm development. The new algorithms introduced in this version were devised by J. Martínez, who also developed the software and generated the products. J. Martínez and C. Gabarró are the main contributor to the writing of this paper. The validation of the products has been carried out by M. Arias, N. Hoareau, M. Umbert, A. Turiel. C. González-Haro. All the coauthors have contributed in the writing and revision of the paper.

*Competing interests.*  The authors declare that they have no conflict of interest.

*Acknowledgements.*  This work has been carried out as part of the ESA Arctic+Salinity project (AO/1-9158/18/I-BG) which permitted to produce the database and the Ministry of Economy and Competitiveness, Spain, through the National RD Plan under L-BAND project ESP2017-89463-C3-1-R. This work represents a contribution to the CSIC Thematic Interdisciplinary Platform PTI Teledetect and PolarCSIC. Argo data were collected and made freely available by the International Argo program and the national programs that contribute to it (https://argo.ucsd.edu, https://www.ocean-ops.org, last access: 1 March 2021). The Argo program is part of the Global Ocean Observing System.



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
