# Peer review of "Improved BEC SMOS Arctic Sea Surface Salinity product v3.1"

_Earth System Science Data, 2021_

## Author Comment (AC1)

**Answers to Reviewer #1:**

This manuscript describes the procedures used to create and validate V3.1 of the BEC Arctic surface salinity product: Arctic+ SMOS SSS v3.1. The description is reasonably complete and well written. The following comments are offered in the spirit of improving the description:

Thanks for the detailed and useful review.

Lines 28-29: "L-band frequency is the region of the electromagnetic spectrum offering the most sensitivity to salinity variations". It is optimum from a remote sensing perspective (protected spectrum and reasonable sensitivity), but the maximum sensitivity occurs at lower frequency (500-900 MHz depending on temperature, incidence angle and polarization).

Yes, we agree with the referee. We have added the following sentence to make it more clear.

"The SMOS frequency band (1.43GHz, L-band) is an optimum band to measure salinity, since this electromagnetic region is protected against human electromagnetic emissions, while the sensitivity to salinity is high."

Moreover, we have adapted the sentence in lines 28-29

Line 51: "available [with] prior registration" ?
 Yes,  corrected.

Line 53: "L1B product contains TB Fourier components": It is not clear in the text whether the starting point is "visibilities" or an image. If starting from the Fourier components, details of the inversion to an image of TB need to be included.

The starting point is the TB Fourier components provided by L1B product. The TB is obtained as the standard procedure does: a Blackman window is used to reduce the Gibbs-like contamination and TB is obtained by an Inverse Fourier Transformation. This text has been included in section 2.1:

"As in the standard procedure  [Anterrieu et al., 2002], we apply a Blackman window to the Fourier components in order to reduce the Gibbs-like contamination. The TB is obtained by applying an Inverse Fourier Transformation to the resulting TB coefficients."
 [Anterrieu et al., 2002]  Anterrieu, E., Waldteufel, P., and Lannes, A. (2002).  Apodization func-tions for 2-D hexagonally sampled synthetic aperture imaging radiometers.IEEE Trans.Geosci. Remote Sens., 40(12):2531–2542.

Line 67: A better reference (better than 2018) for corrections to the Meissner-Wentz model for the dielectric constant of sea water is:  T. Meissner and F. J. Wentz, "The emissivity of the ocean surface between 6 and 90 GHz over a large range of wind

speeds and Earth incidence angles," IEEE Trans. Geosci. Remote Sens., vol. 50, no. 8, pp. 3004–3026, Aug. 2012.

Yes, the reference has been changed.

**Line 117: Why this choice?  For example, how does this compare with the model of Yin et al: "Roughness and foam signature on SMOS-MIRAS brightness temperatures: A semi-theoretical approach," Remote Sens. Environ., vol. 180, pp. 221–233, Jul. 2016.**

The authors state that the roughness model should be improved to be adapted to cold waters. Nevertheless, the choice of the empirical roughness model has not been considered as a part of the improvement algorithm in this approach. The selection of the roughness model has been purely based on previous works performed by this group.

**Line 121: "first Stokes parameter (I = TBx + TBy), parameter used to perform the TB inversion".  Details needed.  For example, how is the roughness correction (which depends on polarization) made?**

The first Stokes parameter is used to avoid ionospheric contribution inaccuracies in the inversion process. The roughness model used is an empirical model based on SMOS measures in which the correction depends on the wind speed and the incidence angle (Guimbard et al 2012). Once all the corrections have been computed (atmospheric, Sun glint, Galactic correction, roughness), we obtain the TB corresponding to the flat sea contribution. The TB is obtained for each latitude/longitude point and antenna position (the antenna position is linked to the incidence angle). The TB inversion is performed minimizing the cost function |I(model)-I(measure)|^2 (a more detailed description is provided in section 2.4 of Martinez et al 2020 -10.13140/RG.2.2.12195.58401)

**Line 152: Typo:  "starting from"**
Done.

**Line 183: See comment 4 above.**
 Reference changed

**Line 184: "conductivity equation Debye (1970)"  The expression attributed to Debye is for the resonance of the water molecule, not conductivity.**

Yes, the correct sentence is *"These dielectric models are based on a Debye relaxation law [Debye, 1929] with a conductivity term."*

The year of the Debye reference has been also modified to refer to the original one and not to a reprint.

**Lines 184-185: "Therefore, we have used the MW model to derive the high latitudes SSS."  This certainly is reasonable, but perhaps it should be noted that the MW model has been shown to result in an SST-dependent bias in the retrieved SSS.**

Yes, it is true. We have developed a bit more the reason for the MW choice mentioning the SST-dependant bias problem

**Line 227: Typo:  measured TB**
 Done

**Line 242-243: "Assuming … high radiometric error …"  There might be other sources of error in addition to noise in the radiometer.**

Yes, of course… for example, RFI is a big problem, however, scenes affected by RFI also have a high radiometric error.

We have added this sentence to the text. '*This procedure will help, also, to mitigate the effect of scenes  contaminated by RFI.'*

**Line 301-302: "It should be noticed the greater coverage and detail of the gradients of Arctic+ v3.1 product to that obtained from the previous BEC Arctic v2.0 product (fig. 5 a-c and 6)."  Wording could be improved.**

We have modified the sentence as follows:  *"Figures 5 and 6 show that Arctic+ v3.1 product has greater coverage and gradient detail than the previous BEC Arctic v2.0 product."*

**Line 318-319: "However, a comparison with punctual measurements can not evaluate the improved data coverage neither spatial resolution."   Something is missing.**

Yes, thanks. Modified by: *"However, a comparison with punctual measurements can not be used to evaluate the improved data coverage nor the spatial resolution."*

**Line 365: Typo: extra series:  should be "to have a long enough series of ..."**

Thanks, changed.

**Line 378: "applied the CTC".  Are there limits on the amount of correlation permitted and how it affects the conclusion?  This could be important since V2 and V3 are so closely related.**

We evaluated the performance of the method as a function of the correlation between the two error-correlated datasets by using synthetic data (see section 3.1 in [Gonzalez-Gambau et al., 2020] ). We defined the following metrics: (i) Fraction of valid retrievals (the ratio of the total valid retrievals to the total number of realizations, (ii) Bias, the difference between the average of all valid estimates of the error standard deviations and the value used for the generation of the dataset and (iii) Uncertainty, the standard deviation of the valid estimates

of error standard deviations. From these experiments, we saw that the dependence of all metrics on the value of the error correlation is weak in most of the cases. Hence, CTC is very robust independently of the degree of correlation between those errors.

**Line 443-445: "As the method extracts the expected natural variability from the common information between the compared products, it means that the fraction of information in the Arctic+ v3.1 product is the largest of the three products." Has this statement about "information" been demonstrated? Perhaps a reference is needed here.**

Yes, the explanation was not correct and not clear enough, and in fact the bullet was repetitive. We have changed the previous bullet as follows:

*"The introduction of the correlated triple collocation also helps to properly assess the differences existing between the current (in 2021) derived satellite products. The metrics show that Arctic+ v3.1 dataset is the one of the three products with the lowest errors in general except in Hudson Bay, east coast of Greenland, and Kara Sea. In particular, the triple collocation shows that SMAP data yields the largest errors."*

**Line 449: "at smaller scales than SMAP and BEC v2.0". With the exception of Fig 13a, this does not appear to be true for SMAP.**

Yes, this comparison explanation between V3.1 and SMAP is missing. We have added this sentence in the spectral analysis section.

*"Moreover, Arctic SSS v3.1 resolves smaller scales than SMAP JPL in Laptev and Bering regions, where SMAP JPL exhibits a flattening in the PDS slope below 50 km wavelength, meaning that the variability contained in SMAP JPL below 50 km wavelength is contaminated by white noise. "*

---

## Author Comment (AC2)

**Answers to Reviewer #2:**

**The paper describes the updated SSS- SMOS derived data set. Inversion algorithm, comparison with other products, including the one previously developed by the same consortium, and partial data validation are described. The paper is clear and the description of process used to move from data to product is appropriate. The main advantage of the new product is its finer resolution which could be relevant for oceanographic processes description. The paper is almost ready for publication except for some minor suggestions which can improve its readability.**

Thank you very much for all your comments and suggestions. We think the paper has improved a lot after integrating them in the manuscript.

**Lines 14-15 : there is a repetition here. Please check.**

Corrected.

**Line 16.19 : you can improve the readability of manuscript here by merging/reformulating the different short sentences.**

The sentence has been changed to:

"The number of in situ surface salinity measurements is, therefore, very scarce, and especially in the central Arctic Ocean, since it is a region with extreme weather conditions and sea ice forces are strong enough to destroy the in situ measurements infrastructures (like Argo floats, moorings, or gliders). "

**Lines 20-21: sentence "…better monitoring the observed changes in the freshwater fluxes". Please add reference for the observed changes which, in my understanding, differs from L-band radiometry.**

Changed to:

"The use of L-band radiometry to fill the observational salinity gaps at high latitudes could be very useful to better monitor the observed changes in the freshwater fluxes (Fournier et al. 2020)."

The reference is also added.

Fournier, S., et al., (2020). Sea surface salinity as a proxy for arctic ocean freshwater changes. JGR: Oceans

**-Line 28-29 : " Whilst L-band frequency is the region of the electromagnetic spectrum offering the most sensitivity to salinity variations, it decreases rapidly in cold waters " this is not true. Recent paper (e.g. 10.1109/TGRS.2021.3101962) proves that other frequencies work better and efforts are made to promote it from space ( DOI: 10.1109/JSTARS.2021.3073286)**

Yes we added the following sentence:

"The SMOS frequency band (1.43GHz, L-band) is an optimum band to measure salinity, since this electromagnetic region is protected against human electromagnetic emissions, while the sensitivity to salinity is high."

And modified the cited one by this one:

"Whilst the sensitivity to salinity is high at L-band, the sensitivity decreases rapidly in cold waters."

**-Line 31-32 – LSC: you can simply mention that the problem is due to the large footprint on the ground (interferometer is obviously worst).**

We have changed the sentence accordingly.

**-Line 58 : could you provide an estimate of error introduced by the interpolation process?**

This is not possible in this case. The reliability of the nearest-neighbor interpolation depends on the spatial variability of the original ECMWF data.

**-Line 67 – the sentence "The SSS and SST are converted to TB "sounds a bit strange, I guess that you meant that SSS and SST are used as inputs in an e.m model to generate simulated Tb values.**

Yes, the referee is right. The sentence has been changed.

"*The SSS and SST are used as inputs of the Meissner and Wentz dielectric constant model to obtain TB (Meissner & Wentz, 2004, 2012). The TB obtained is considered the reference value to perform the spatial bias correction of the measured TB.*"

**-Line 69_ you mention that data "generated from measurements of the 2005-2017 period" are used as reference but SMOS data refers to 2011-2019 period. Did you evaluate if there is an impact on the obtained the results if the same overlapping period is used?**

It is not possible to have the same overlapping period because WOA is generated only for the complete 2005-2017 period while SMOS has no data prior to 2010 (2011 in our case).

**-Line 90: could you add an accuracy estimate for Tara data?**

The Tara expedition in the Arctic used a thermosalinograph (TSG) Seabird SB45 to measure the conductivity and temperature and then the salinity is computed. This sensor has an accuracy of the conductivity of ± 0.0003 S/m, which represents the same magnitude of salinity accuracy of 0.0003 PSU.

Reference: https://www.seabird.com/sbe45-microtsg-thermosalinograph/product?id=54627900541

**-Line 104-105: could you estimate/quantifying the differences in considering 64x64 instead of 128x128 point? You mention "without loss of information/resolution ".**

A division of the antenna hexagonal grid in 64x64 cells provides 4096 grid points. This is enough to provide the Tb values because the number of visibilities from which snapshots are derived by a linear transformation is 2791. The hexagonal grid must be constructed as 2nx2n grid and n<6 undersamples the image. This fact has been explained in the new text:

*"This resolution in the antenna level results in 4096 grid points being enough to provide the TB values because the number of visibilities from which snapshots are derived by a linear transformation is 2791. This choice allows us to reduce the computational time without loss of information/resolution."*

**-Line 119-120; is not clear if the ionosphere correction is applied. Since 1st Stokes parameter is used for the inversion**

Yes. It is applied. This contribution is accounted for in figure 1.

**-Line 133-135: if I understood correctly you use, as reference, the SSS value obtained from the WOA instead to the one obtained from SMOS (average value). If so, which is the estimated differences between these values? Could you provide an example for some specific regions where the coast contamination is /or isn't relevant?**

In this new version of the Arctic salinity product, the bias is corrected in TB and not in SSS. In this case the reference is the TB obtained from SSS and SST from WOA. The reference TB is computed from WOA SSS and SST using the Meissner and Wentz dielectric model. Then, we obtain TB values for each latitude, longitude, satellite orientation and position in the antenna reference frame (the new reference). The correction depends of a wide variety of factors but can attain values as high as 10K depending on the position in the antenna reference frame or the distance to the coast. Figures 19 and 20 from ATBD document (http://dx.doi.org/10.13140/RG.2.2.12195.58401) show the differences for 4 points of the field of view for ascending and descending cases (reproduced here).

[Figure]

**Figure 19:** Correction to be applied to measured half first Stokes parameter in 4 difference FOV positions for ascending passes (equation 20)

[Figure]

**Figure 20:** Correction to be applied to measured half first Stokes parameter in 4 difference FOV positions for descending passes (equation 20)

-Line 173 : the sentence "Only latitudes above 50⁰N are considered" can be eliminated since it is repetition of line 159.

The sentence has been eliminated.

-Line 189- For the minimization did you use different approaches? For instance, did you check if the introduction of a regularization term could be beneficial?

The minimization used is a non bayesian method and therefore it doesn' have any regularization term. We haven't tested any other minimization algorithm for this Arctic product.

-Line 203 – 206 why using 100 , 7 and 2 as criteria? Could you better justify it? (i.e. Why not 90 or 110?)

The number of the minimum number of measures to create the SMOS-based climatology was taken to 100 by simply testing different values. No significant differences are obtained using 90 or 110… The value is based on the minimum required measures to obtain a statistically significant TB distribution

without holes. The resulting distributions should be adequate to obtain their moments and this is not reliable with a low number of measures. Moreover, points with a low number of TB measures in 9 years are not certainly reliables.

The values adopted for the maximum absolute kurtosis (7) and absolute skewness (2) may be used as reference values for determining substantial non-normality of the distribution. West et al 1995, recommend these values as the values for which the distribution begins to depart substantially from normality.

[West et al (1995)] West SG, Finch JF, Curran PJ. Structural equation models with nonnormal variables: problems and remedies. In: Hoyle RH, editor. Structural equation modeling: Concepts, issues and applications. Newbery Park, CA: Sage; 1995. pp. 56–75.

**-Line 230-233: Also here : could you better explain how the thresholds were selected?**

These thresholds have been selected purely by trial and error. However, they are not too restrictive: salinity must be positive and less than 50 psu, for usual salinity values we don't expect large differences between the retrieved value and WOA value, and for low salinity values we relax the value to take into account possible rivers discharges and ice melting (WOA does not account for transient states)

**-line 260: The 12 psu bias has, in my view, severe implication. It implies that the retrieval largely overestimates the retrieved SSS besides the numerous procedures, averaging, de-bias which were conducted to derive it. Did you have an explanation for the bias? There is a problem of representativeness of SSS retrieved by SMOS and what provided by HYCOM? Or there is an absolute error on HYCOM only as bias but it didn't affect the temporal variability? At the end you plan to use it for the temporal correction.**

There is no problem with HYCOM (or at least its use in the temporal correction is not the cause of this bias). The main cause is that WOA2018 is assuming a Gaussian distribution of SST and SSS whereas the first stokes distributions provided by SMOS are generally positively skewed (a more detailed explanation is provided in section 2.7 of the manuscript "Correction of the residual spatial bias"). The temporal correction is performed before the correction of residual spatial bias due to computational optimization requirements, requiring this high initial step to speed up the convergence.

**-line 276: how much the number differs from zero? This could be a useful information for the reader.**

The weighted average of all L2B orbits in the 2013-2019 period minus the value provided by WOA ranges between -10 and 10 psu but mainly between -2 and 2 psu. The values differ a lot between the

Arctic zones being smaller in the open ocean and larger in the North Sea (negative), in the Beaufort Sea (positive) and the East Siberian Sea (positive) . [See following figures]

This information, not the figures, has been included in the text.

[Figure]

**Figure 28:** Weighted average of all ascending L2B orbits in the period 2013-2019 minus WOA18-A5B7

**Figure 29:** Weighted average of all descending L2B orbits in the period 2013-2019 minus WOA18-A5B7

**-figure 5: why the error is only represented for radiometric uncertainty? you have different factors that contribute to the error computation. This is bit reductive**

Yes, we agree that this is only a portion of the error, but the radiometric error is the only one that can be computed with an acceptable degree of reliability. Moreover, it is the only one that depends on the instrument and not on the geophysical models.

**-Argo validation: not clear to me why BEC V2.0 data provides better results in 2011 -2012. If SMOS is affected by RFI (as the authors mention) this impact on all the products. Moreover, for BECV3.1 you develop an approach which is devoted to mitigate RFI effects.**

This is mainly because in v2.0 the bias was corrected using ARGO data. Therefore, it is expected a better correspondence with ARGO data. However, in v3.1 the approach was not focus on mitigating RFI effects, but to improve the inversion at low SSS values, and therefore improve the spatial resolution.

**-Tara Validation: from table it seems to me that the affirmation "Arctic+ v3.1 product, better than the previous BEC v2.0 product in most of the seas" is questionable. I see a clear advantage in two cases only (Kara and Beaufort).**

Yes, the reviewer is right! We have specified on the text when v3.1 better than v2.0 with this sentence.

"Matchups with TARA are different results depending on the sub-basin. Arctic+ v3.1 product have less RMSD than BEC v2.0 product for three sub-basins (Kara, East Siberia and Beaufort seas) and also for the global value."

**-Spectral Analysis : the difference between SMAP and BECV3.1 seems to me very small in the figure then my conclusion is that both method provide similar results. It is correct?**

We have added this sentence to clarify:

"Moreover, Arctic SSS v3.1 resolves smaller scales than SMAP JPL in Laptev and Bering regions, where SMAP JPL exhibits a flattening in the PDS slope below 50 km wavelength."

**-Lie 427: while I recognize that validation for 3 days product require additional effort, although it could be very interesting for potential users, I believe that validation at 18 days should be simple and can be easily implemented.**

We agree, that the validation of the two additional products (3 days and 19 days) would be interesting, but this is out of scope of the project Arctic + Salinity and of this paper. However, this is something we have in our plans to do.

---

## Author Comment (AC4)

**Improved BEC SMOS Arctic Sea Surface Salinity product v3.1**

Justino Martínez1, Carolina Gabarró1, Antonio Turiel1, Verónica González-Gambau1, Marta Umbert1, Nina Hoareau1, Cristina González-Haro1, Estrella Olmedo1, Manuel Arias2, Rafael Catany2, Laurent Bertino3, Roshin Raj3, Jiping Xie3, Roberto Sabia4, and Diego Fernandez5 1 
[revised manuscript text omitted]